# Economic Impact of the Implementation of an Enhanced Recovery after Surgery (ERAS) Protocol in a Bariatric Patient Undergoing a Roux-En-Y Gastric Bypass

**DOI:** 10.3390/ijerph192214946

**Published:** 2022-11-13

**Authors:** Alfonso Higueras, Gilberto Gonzalez, Maria de Lourdes Bolaños, Maria Victoria Redondo, Isabel M. Olazabal, Jaime Ruiz-Tovar

**Affiliations:** 1School of Medicine, University Alfonso X, 28691 Madrid, Spain; 2Hospital Real San José, Guadalajara 19001, Mexico; 3Neuroscience Institute, Centro Universitario de Ciencias Biológico Agropecuarias (CUCBA), University of Guadalajara, Guadalajara 44600, Mexico

**Keywords:** Roux-en-Y gastric bypass, economic analysis, pharmacological costs, surgical material costs, surgical time costs, complementary test costs, bed occupancy costs

## Abstract

Introduction: Enhanced Recovery After Surgery (ERAS) protocols have proven to be cost-effective in various surgical procedures, mainly in colorectal surgeries. However, there is still little scientific evidence evaluating the economic impact of their application in bariatric surgery. The present study aimed to compare the economic cost of performing a laparoscopic Roux-en-Y gastric bypass following an ERAS protocol, with the costs of following a standard-of-care protocol. Patients and methods: A prospective non-randomized study of patients undergoing Roux-en-Y gastric bypass was performed. Patients were divided into two groups: patients following an ERAS protocol and patients following a standard-of-care protocol. The total costs of the procedure were subdivided into pharmacological expenditures, surgical material, and time expenses, the price of complementary tests performed during the hospital stay, and costs related to the hospital stay. Results: The 84 patients included 58 women (69%) and 26 men (31%) with a mean age of 44.3 ± 11.6 years. There were no significant differences in age, gender, and distribution of comorbidities between groups. Postoperative pain, nausea or vomiting, and hospital stay were significantly lower within the ERAS group. The pharmacological expenditures, the price of complementary tests performed during the hospital stay, and the costs related to the hospital stay, were significantly lower in the ERAS group. There were no significant differences in the surgical material and surgical time costs between groups. Globally, the total cost of the procedure was significantly lower in the ERAS group with a mean saving of 1458.62$ per patient. The implementation of an ERAS protocol implied a mean saving of 21.25% of the total cost of the procedure. Conclusions: The implementation of an ERAS protocol significantly reduces the perioperative cost of Roux-en-Y gastric bypass.

## 1. Introduction

Recent advances in the perioperative care of patients undergoing bariatric surgery, optimization of operative techniques, improvements in surgical devices, and standardization of bariatric surgery programs have resulted in a decrease in morbidity and mortality of these procedures. Laparoscopic approaches are being performed with increasing frequency, with laparoscopic Roux-en-Y gastric bypass (RYGB) and laparoscopic sleeve gastrectomy being the most common bariatric techniques [1]. Given the progressive increase in obesity and related diseases and the consequent increase in the economic cost associated with their treatment, the current challenge is to increase the cost-effectiveness of bariatric surgery while maintaining the current low rates of associated morbidity in these patients [1,2].

Operating theatre occupancy time, medical staff, and hospital stays are expensive, limited, and not available indefinitely. Optimizing the best logistics and utilization of resources would increase both production and quality of care [3].

The first Enhanced Recovery After Surgery (ERAS) protocol was developed by Kehlet in 1997. ERAS protocols are logistical programs with abundant scientific evidence in their favor, primarily in colorectal surgery. They have demonstrated that an “evidence-based” approach to perioperative care leads to earlier postoperative recovery and shorter hospital stay, with an increased sense of well-being for the patient. Although the content of different ERAS programs may vary depending on the surgical procedure to be performed, there are factors common to all. These factors include the use of minimally invasive surgical procedures, the introduction of short-acting anesthetic agents, optimized pain and nausea/vomiting control, early oral nutrition, and ambulation. The aim of all these measures is to reduce the body’s perioperative stress response, allowing early restoration of organ function [4,5,6].

The essence of these programs is multimodal and multidisciplinary management, and many authors have already demonstrated the safety and feasibility of their application to bariatric surgery [7,8].

Based on this concept, the ERAS Spain group developed a multidisciplinary ERAS program for bariatric surgery. The implementation of this protocol has proven to be safe, with a high compliance rate for most items [9]. When compared with a historical cohort of standard care, the Spanish ERAS protocol for bariatric surgery demonstrated similar outcomes to standard care in terms of complications, re-interventions, mortality, and readmissions. However, it was associated with less postoperative pain, lower levels of acute phase reactants, and earlier hospital discharge [10].

Our group conducted the first prospective randomized clinical trial in 2019, comparing the implementation of an ERAS protocol with a standard care protocol in RYGB. This study concluded that the implementation of the ERAS protocol was associated with less postoperative pain, nausea and vomiting, lower levels of analytical acute phase reactants, and earlier hospital discharge, but with similar rates of complications, reoperation, mortality, and readmission [11].

The economic impact of applying ERAS protocols has been widely described in colorectal surgery. Sammour et al. were the first to demonstrate savings per patient of 4429$ [12]. Lee et al. first and Lemanu et al. later published two systematic reviews, concluding that the ERAS protocol was a cost-effective intervention in colorectal surgery, with associated healthcare cost savings [13,14]. Feng et al. studied the application of ERAS protocols applied to laparoscopic colorectal approaches, observing that they reduced hospital costs due to improved recovery and reduced hospital stay [15]. Given the current evidence, many institutions are opting to systematically introduce ERAS protocols, without limiting it to colorectal surgery, but extending it to other surgical areas. 

In bariatric surgery, simply the reduction of hospital stay seems to be a sufficient economic argument in favour of the implementation of ERAS protocols. In fact, the first Enhanced Recovery After Bariatric Surgery (ERABS) guidelines were already published in 2016 [16], and have been recently updated [17].

The present study aimed to compare the economic cost of performing a laparoscopic Roux-en-Y gastric bypass following an ERAS protocol, with the costs of following a standard-of-care protocol. 

## 2. Materials and Methods 

A prospective non-randomized study of patients undergoing RYGB was performed between March and December 2021. Inclusion criteria were body mass index (BMI) > 40 kg/m^2^ or >35 kg/m^2^ with the presence of comorbidities associated with obesity. Exclusion criteria were patients undergoing other bariatric techniques, underlying severe cardiovascular disease, chronic renal failure, liver dysfunction, previous foregut surgery, active infectious or neoplastic disease, unacceptable surgical risk, addiction to drugs or alcohol, endocrine diseases causing obesity, severe psychiatric disease, and any other contraindication to bariatric surgery.

In the absence of conclusive data on the possible reduction of costs after the implementation of an ERAS protocol, we chose to determine the sample size based on postoperative pain quantified by a visual analog scale (VAS) 24 h after the intervention. An eventual reduction in postoperative pain reduces the need for extra administration of analgesics, mainly opioids, which results in delayed mobilization, a longer postoperative ileus, a delay in oral food tolerance, and therefore a longer hospital stay.

Based on a previous study of our group [11], in which pain 24 h after surgery was quantified as 37 mm in patients who underwent standardized perioperative care and 16 mm in patients who complied with an ERAS protocol, and at 80% power and a significance level of *p* < 0.05, it was calculated that 42 patients needed to be included in each study group. This pain quantification was measured by visual analogic scale (VAS), ranging from 0 (absence of pain) to 100 (unbearable pain).

To comply with the ERAS protocol, certain devices were required, the availability of which could not be assured by the supplier for the duration of the study. For this reason, it was decided not to randomize the patients, and the assignment to each group was decided according to the availability of the necessary devices at each moment until the established sample size was met. The study was blinded to the outcome evaluators.

### 2.1. Preoperative Evaluation

In both groups, candidates for bariatric surgery were evaluated by a multidisciplinary team consisting of surgeons, endocrinologists, nutritionists, anaesthesiologists, and psychiatrists. Preoperative assessment included abdominal ultrasound, upper gastrointestinal endoscopy, polysomnography, and blood tests. Interviews and psychological surveys were conducted to assess patient involvement in following an adequate diet after surgery. 

The nutritionist established a balanced preoperative diet three months before surgery, with a restriction of 300 Kcal/day on the basal metabolic expenditure, calculated individually using the Harris-Benedict formula [18], to achieve a preoperative weight loss of between 5–10% of the total weight.

### 2.2. Surgical Technique

Five ports are inserted (three 12 mm and two 5 mm atraumatic ports). The procedure begins by freeing the angle of Hiss until the left diaphragmatic crura is identified using a harmonic scalpel. A gastric pouch of about 12 cm in length is made: a first horizontal cut is made using an EndoGIA (EndoGIA, Medtronic, Minneapolis, MI, USA). Next, a 36 Fr Foucher orogastric tube is inserted and three vertical cuts are made with a 60 mm Endostapler with 60 mm beige cartridges until the gastric reservoir is completely freed from the stomach remnant.

The greater omentum is divided with a harmonic scalpel, the Treitz angle is identified and the length of the bowel is counted, leaving 70 cm of biliopancreatic limb and 150 cm of the alimentary limb. The anastomosis between the gastric pouch and the alimentary limb is performed with Endostapler and a 30 mm blue cartridge. The remaining orifice is closed with a 2/0 barbed suture (V-Loc, Medtronic, Minneapolis, MI, USA). Entero-enteral anastomosis between the biliopancreatic and alimentary limbs to create the common limb (Roux-en-Y) is performed with Endostapler and a 45 mm beige cartridge. The remainder of the orifice is closed with a 3/0 barbed suture. The tightness of both anastomoses is checked with methylene blue dye. The ports are removed under direct vision. No aponeurotic suturing of the port orifices is performed.

### 2.3. ERAS Protocol

The group in which the ERAS protocol was applied followed the Spanish national protocol, developed and approved by the Spanish ERAS Group (GERM) and validated in a multicenter pilot study [9]. The complete protocol is described in Table 1.

Two weeks before surgery, nutritional hypocaloric hyperproteic formulas were prescribed to maximize the advisable preoperative weight loss of at least 10% of the patient’s weight. Patients were informed that this weight loss was beneficial and could allow a better recovery within the ERAS protocol.

During surgery, goal-guided fluid therapy was applied, which was determined by indirect methods of central venous pressure, such as the ClearSight device (Edwards Lifesciences, Irvine, CA, USA), which establishes an estimate of central venous pressure through capillary refill.

No central venous catheter or urinary catheter were routinely inserted. The protocol applied for the management of postoperative nausea or vomiting (PONV) followed the consensus recommendations of the Society for Ambulatory Anesthesia [19]. All patients undergoing bariatric surgery were considered to be at high risk of PONV and therefore triple antiemetic prophylaxis was administered, including dexamethasone during anaesthetic induction and droperidol and ondansetron at the end of surgery. Nursing recorded the presence of PONV every 6 h. If PONV occurred, metoclopramide 10 mg intravenously was used as postoperative treatment. No intra-abdominal drains or nasogastric tubes were used in the ERAS protocol. Early mobilisation and oral fluids administration were performed 6 h after surgery.

Multimodal analgesia consisted of a laparoscopically guided transverse abdominal muscle plane (TAP) block. It consisted of an infiltration with 0.25% bupivacaine in the space between the internal oblique and transverse abdominal muscles, through which the abdominal wall nerve trunks pass. The infiltration was performed bilaterally, at the level of the trocar insertion points, but lateral to them.

Postoperative analgesia included the administration of metamizole 2 g/8 h and paracetamol 1 g/8 h, alternating every 4 h. When postoperative pain, as measured by VAS, overcame 50 mm at any time during the postoperative course, 5 mg of morphine was administered subcutaneously.

Patients were fully informed of all steps of the ERAS protocol, including the need for preoperative weight loss, a preoperative fasting period of only 6 h for solids and 2 h for liquids, and early postoperative oral intake and mobilization. They were also informed that, if they had no complications, they would probably be discharged the first day after surgery, provided that pain was controlled with oral analgesia, full ambulation was achieved, and the patient accepted the discharge. It was explained to the patients that this early discharge was a safe act, that they would receive a telephone call from the nursing staff to monitor their condition, and that they should go to the outpatient clinic 2 weeks after surgery for medical review and analytical control. Before discharge, they also received nutritional education for the postoperative course by the physician and nursing staff, for wound care and physical activity. Patients were discharged following the criteria established by the protocol.

### 2.4. Standard Care Protocol

The main differences between this standard care protocol and the ERAS protocol were:-Preoperative fasting of 12 h for both solids and fluids.-Routine bladder catheter placement-Fluid therapy estimated by patient’s weight-Intraoperative analgesia with Remifentanil, without minimizing the use of opioids.-Prophylaxis of nausea and vomiting with Ondansetron and Dexamethasone-Placement of intra-abdominal drainage-Intraoperative analgesia with nonsteroidal anti-inflammatory drugs and opioids, without multimodal analgesia methods-Bed rest for 24 h postoperatively-Initiation of oral fluid tolerance 24 h after surgery.

### 2.5. Determination of Costs

The total costs of the procedure were determined and subdivided into pharmacological costs (including all drugs administered during the pre-, intra-, and postoperative period), surgical material costs (including both surgical and anesthetic devices), costs of complementary tests performed during the hospital stay, and costs related to the hospital stay (including the cost of using a standard hospital bed as well as an ICU bed). The costs were calculated based on the value assigned to each unit in the hospital where the interventions were conducted.

### 2.6. Variables

The recorded variables include demographic data, comorbidities, anthropometric measures, morbidity, mortality, hospital stay, and readmission. PONV was also assessed. The postoperative pain score, as measured by VAS 24 h after surgery, was determined. Morphine rescue needs during the first 24 h postoperatively were also analyzed. Postoperative pain and PONV during the first 24 h after surgery was assessed by a nurse blinded to the applied protocol. Hospital discharge was decided by a surgeon blinded to the treatment, whose decision was based on the common discharge criteria in both protocols.

The total costs of the procedure were calculated and analyzed separately the pharmacological costs, the costs associated with the surgical procedure, the costs of in-hospital complementary tests, and the costs associated with the hospital stay. The economic costs were calculated according to the scales established for the center where the surgeries were performed.

### 2.7. Statistical Analysis

Quantitative variables following normal distribution were defined by mean, standard deviation, and range of values. For those variables that did not follow a Gaussian distribution, the median was used instead of the mean as a measure of centralization. Discrete variables were defined by the number of cases and percentage.

For the analytical study of the variables we used:-Comparison between qualitative variables: In the case of comparing two discrete variables, the Chi-Square test was used. When the expected value was less than 5 in any of the boxes of the table, it was necessary to use Fisher’s exact test. The magnitude of the association was estimated using the Odds Ratio.-Comparison of two independent means: Student’s *t*-test (Mann Whitney U test for non-Gaussian variables).

Data processing and analysis were carried out with the statistical software SPSS 22.0 for Windows. Values of *p* < 0.05 were considered statistically significant.

## 3. Results

The 84 patients included 58 women (69%) and 26 men (31%) with a mean age of 44.3 ± 11.6 years, ranging from 22 to 68 years. There were no significant differences in age and gender distribution between groups. The distribution of pathologies within the personal history is summarized in Table 2. Most diabetic patients were on oral anti-diabetic treatment, with no significant differences between groups. All hypertensive patients were under pharmacological treatment. Patients diagnosed with SAHS were using CPAP, and all of them require at least 6 weeks of adaptation to the device before surgery.

Anthropometric variables at the first visit to the Outpatient Clinic and weight loss obtained with the balanced preoperative diet 3 months before surgery are summarized in Table 3. No significant differences were observed between groups. In the ERAS group, nutritional hypocaloric hyperproteic formulas were prescribed during the last 2 weeks before surgery, increasing their weight loss up to 12.9 ± 3.4% of total weight.

All the patients underwent a Roux-en-Y gastric bypass without any case of conversion to laparotomy. The mean operative time was 94.2 ± 18.5 min in the standard protocol group and 98.7 ± 21.5 min in the ERAS group (*p* = 0.305).

In both groups, the complication rate was 2.4% (1 patient in each group), consisting of one case of haemoperitoneum in the ERAS group and one case of anastomotic leak in the standard protocol group. Reoperation and a 24-h postoperative ICU stay were required in both groups. There was no mortality in either group.

### 3.1. Hospital Stay

The mean hospital stay was 2.8 ± 1 days (Range 2–7 days) in the standard protocol group versus 1.4 ± 0.8 days (Range 1–5 days) in the ERAS group (Mean difference 1.4 days (95% CI (1–1.8); *p* < 0.001). In the standard protocol group, no patient could be discharged on the first postoperative day, while in the ERAS group 29 patients (69%) were discharged within 24 h of surgery (RR 3.2; 95% CI (2.1–5.1); *p* < 0.001). No patient in the study required readmission to the hospital.

### 3.2. Postoperative Nausea or Vomits

The rate of nausea or vomiting in the ERAS group was 4.8% compared to 21.4% in the standard protocol group (RR 0.183; 95% CI (0.034–0.903); *p* = 0.024).

### 3.3. Postoperative Pain

Mean postoperative pain values at 6 and 24 h postoperatively, quantified by visual analog scale (VAS), are summarised in Table 4. At 6 h postoperatively, the standard protocol group had a higher mean pain of 20.5 mm (95% CI 11.4–29.5; *p* < 0.001). At 24 h postoperatively, the standard protocol group also had a higher mean pain of 10.2 mm (95% CI 2.8–17.7; *p* = 0.008).

Twelve patients (28.6%) in the standard protocol group required analgesic rescue with morphine chloride, of which four required rescue on two occasions during their hospital stay. However, only one patient (2.4%) in the ERAS group required analgesic rescue with morphine chloride (RR 11; CI95% (3.6–22.4); *p* < 0.001).

### 3.4. Cost Analysis

#### 3.4.1. Pharmacological Cost

The mean pharmacological cost in the standard protocol group was 356.1 ± 16.7$ versus 337.6 ± 14.9$ in the ERAS group (*p* < 0.001).

Although a priori extra drugs are used for multimodal analgesia and intense antiemetic prophylaxis according to the Apfel scale, the pharmacological cost is significantly higher in the standard protocol group, due to the greater need for drugs to treat complications (pain, nausea, vomiting, …).

#### 3.4.2. Cost of Surgical Time and Surgical Material

The mean cost of surgical time and surgical material in the standard care group was 3257.67 ± 277.6$ versus 3270.69 ± 322.9$ in the ERAS group (NS).

#### 3.4.3. Cost of In-Hospital Complementary Tests

The mean cost of in-hospital complementary tests in the standard care group was 423.4 ± 58.6$ versus 311.5 ± 39.6$ in the ERAS group (*p* < 0.001). Although the rate of complications was similar in both groups, in the standard care group a higher number of blood tests, abdominal X-plain films, and abdominopelvic CT scans were performed in the presence of nausea and vomiting or increased pain, to rule out postoperative complications.

#### 3.4.4. Cost of Hospital Stay (Conventional and ICU Bed Occupancy)

The cost of hospital stay in the standard care group was 2828.3 ± 1429.2$ versus 1466.7 ± 1201.6$ in the ERAS group (*p* < 0.001).

#### 3.4.5. Total Cost of the Procedure

Adding up the different cost items, the total cost of the procedure was significantly higher in the standard care group (mean difference 1458.62$ per patient (95% CI (851.16–2066.07; *p* < 0.001). The implementation of an ERAS protocol implied a mean saving of 21.25% of the total cost of the procedure.

The distribution of costs between groups is summarized in Table 5.

## 4. Discussion

In the last decades, the number of surgical procedures has significantly increased [20], making it essential to optimize processes and techniques [21].

The ever-increasing health needs of patients [22], mainly due to the increasing prevalence of chronic diseases and the cost of their treatment, are critical variables in the medical procedures to be implemented. It is therefore urgent to introduce new methods to ensure the sustainability of routine surgical practice in the current scenario, characterized by the scarcity of healthcare resources [2].

Multiple institutions are choosing to systematically introduce ERAS protocols based on the scientific evidence showing patient benefits. Following the publication of a study by Sammour et al. [23], there is sufficient evidence to support the hypothesis that the initial burden and cost of implementing an ERAS program are offset by superior savings. Sammour et al. showed savings per patient of 4429$ when an ERAS protocol was applied to colorectal surgery. The estimated cost of eventually avoidable complications when applying an ERAS protocol in colorectal surgery was estimated by Birkmeyer et al. to be up to 10,000$ per patient [24]. However, the different studies reporting cost savings have shown great variability among them, even when referring to the same surgical approaches [13,14]. Nevertheless, all studies agree that the cost reduction is due to improved postoperative recovery and reduced hospital stay [15].

Similar cost savings have been demonstrated among other surgical procedures, such as hepatobiliopancreatic surgery [25], urologic surgery [26], gynecologic [27], and cardiovascular procedures [28]. Referring to gynaecologic surgery, in 2019 Pache et al. found out that the main costs of implementing an ERAS protocol were specialized staff, administration times, consumables, and investment in specific auditable software. They estimated the average overcharge per patient as approximately 687$. However, subsequent savings were estimated at 4381$ per patient [29]. All these cost overruns of program implementation are significantly reduced once the protocol is established and included within routine clinical practice.

To our knowledge, this is the first study evaluating the economic impact of the implementation of an ERAS protocol on Roux-en-Y gastric bypass as a primary bariatric procedure. Many bariatric surgeons have been reluctant to admit the implementation of an ERAS protocol, arguing that current bariatric surgery already involves in most cases a minimally invasive approach, which undoubtedly improves postoperative recovery and reduces hospital stay. However, as mentioned above, ERAS protocols are multidisciplinary and do not only consist of the application of one item. Therefore, we consider it essential to conduct a prospective study comparing the implementation of a complete ERAS protocol versus a standard care protocol, which already includes the minimally invasive approach, and to analyze not only the clinical outcomes but also its economic impact.

At the clinical level, our study confirms what has been described in our previous pilot studies [9,10], as well as in other studies by other authors [1,3,8]: by applying the ERAS protocol, no differences were observed in the incidence of complications, mortality, and readmissions, but a shorter hospital stay was determined. In addition, in the present study, we consider two clinical aspects involved in achieving a shorter hospital stay among the ERAS protocol patients, such as less nausea and vomiting and less pain perception. Although not considered in this study, both aspects would be directly related to a better quality of recovery.

Referring to the economic aspects, we have subdivided the calculation into pharmacological cost, cost of surgical material and time, cost of complementary tests, and cost of bed occupancy (conventional and ICU). According to the study of Pache et al. [29], a higher pharmacological cost would have been expected (due to the administration of a greater number of prophylactic drugs: antiemetics, multimodal analgesia, etc.) and a higher cost associated with the use of intraoperative devices and operating room occupancy time. However, the pharmacological cost in our study was significantly lower in the ERAS group. Studying this aspect on a case-by-case basis, we observed that the cost of the drugs used to treat adverse effects (analgesics, antiemetics, etc.) was higher than the prophylactic use of these drugs.

Regarding operative time and surgical material used, it is true that it was slightly more expensive in the ERAS group, without reaching statistically significant differences. The use of certain specific devices makes the procedure more expensive, and the use of multimodal analgesia slightly prolongs the operative time. However, the non-use of a central venous catheter, bladder catheter or drains also meant savings for the ERAS group, and offset the cost overruns by other devices employed.

Another relevant point to consider is the cost related to complementary tests. The implementation of an ERAS protocol in bariatric surgery does not imply the performance of extraordinary tests. On the contrary, an increase in the incidence of nausea and vomiting, or worse pain control makes the surgeon suspect the presence of a postoperative complication and induces him to request unnecessary blood analysis or imaging tests.

The greatest differences in sanitary costs are seen in hospital stays, where compliance with an ERAS protocol almost halves the cost attributable to this item. In the absence of fewer complications, this reduction in hospital stay does not seem justifiable, and some believe that it is exclusively due to forcing earlier discharge in these patients. Firstly, the preoperative information given to the patient on the different points of the protocol, including early discharge in the absence of complications, allows the patient to understand his situation and accept discharge after 24 h, having explained to him the warning signs and symptoms for which he should consult a doctor [9,10,11].

In this study, hospital discharge was achieved within 24 h of surgery in 69% of patients in the ERAS group, compared to no patients in the other group, in the absence of a higher number of complications. This indicates not only the importance of patient information but also that more factors influence patient well-being and enable patients to accept early discharge. These are mainly the correct tolerance to oral feeding, in the absence of nausea or vomiting, and adequate pain control with oral non-opioid analgesics. However, future studies should include quality questionnaires during the immediate postoperative period to confirm these hypotheses.

The present study implores the idea of ERAS protocol for bariatric surgery in the setting of a developing to a developed economy. However, the idea of reduced cost and early recovery are even more imperative for countries which are already strained by a low to meagre healthcare budget and at the same time have higher incidences of obesity disorders. The current evidence is based on studies carried out in developed countries, but given this clear economic benefit, it is logical to think that its application in developing countries would also be a cost-effective measure. It is true that the availability of some complex technological devices may be more difficult in this environment, but even without them, many of the items in the ERAS protocols could be implemented without problems, and a clear benefit could be obtained. Further studies should evaluate this hypothesis.

Finally, we would like to highlight as the main limitation of this study, the impossibility of randomizing patients due to the unavailability of Clearsight devices, which would allow the calculation of goal-directed intraoperative fluid therapy for some time during the study. Goal-directed fluid therapy is considered an essential part of our ERAS protocol, as it optimizes fluid infusion, thereby reducing bowel loop edema, which contributes to postoperative ileus and subsequent nausea and vomiting. Therefore, we preferred not to randomize the study rather than remove this point from the protocol.

Another potential limitation of the study is that the health-care costs associated with the procedure have been established for a morbidly obese population with homogeneous characteristics between the two study groups. However, populations with a higher prevalence of comorbidities associated with obesity would have possibly presented higher rates of postoperative complications and would even have made the implementation of the ERAS protocol more difficult, resulting in delayed hospital discharges. For instance, in our study only the prevalence of diabetes mellitus diagnoses was recorded, whereas baseline glucose, insulin levels, and HOMA-IR of the patients as an index for insulin resistance would have added more information and could even be markers of patients more prone to develop complications or presenting difficulties in the adherence to the ERAS protocol. Body composition, as measured by bioimpedancy, including lean and fat mass percentages would have also added valuable information. Therefore, extrapolation of the results of the present study to any type of patient should be taken with caution.

## 5. Conclusions

The implementation of an ERAS protocol significantly reduces the perioperative cost of Roux-en-Y gastric bypass as a primary bariatric procedure. There were no significant differences in complication or readmission rates between groups. However, patients in the ERAS group had less postoperative pain and less nausea and vomiting, which resulted in less need for rescue medication for their treatment. Therefore, the reduction in the economic cost of the process was not only due to a shorter hospital stay but also to the reduced need to administer medication and to perform complementary tests to rule out complications in the presence of pain or vomiting, which were ultimately the determining factors in prolonging the length of hospitalization in patients in the standard protocol group.

## Figures and Tables

**Table 1 ijerph-19-14946-t001:** Spanish National ERAS protocol for bariatric surgery [9].

Time	Components of the Protocol
Preoperative	- Provision of verbal and written information to patients regarding the ERAS. Collection of signed consent. - Preoperative evaluation: nutritional, cardiologic, anemia, and comorbidity optimization, if required. - Laboratory data: glycemic, lipid, hepatic, and iron profiles; basal arterial gasometry endocrinologic assessment. - Polysomnographic study to control and/or diagnosis of SAHS; start CPAP at least 4–6 weeks before surgery. - Hypocaloric diet 3 months before surgery; nutritional hypocaloric hyperproteic supplementation 2 weeks before surgery. A 10% total weight loss before intervention is advised.
Day before surgery	- Low-residue diet - Thromboprophylaxis - Fasting 6 h solid; 2 h clear liquid - Avoid anxiolytic drugs
Perioperative	Preoperative: - Placement of compression stockings or intermittent pneumatic compression according to thromboembolic risk. - Peripherical catheter placement - Antibiotic prophylaxis 1 h before surgical incision. Intraoperative Administration of antireflux prophylaxis (metoclopramide + ranitidine 30 min before anesthetic induction) Usual measures for orotracheal intubation in patients with difficult airway; rapid sequence orotracheal intubation Alveolar recruitment maneuvers after orotracheal intubation Maintenance: oxygen/air with FiO2 60–80% Hemodynamic optimization: goal-directed fluid administration (Clearsight, Edwards Lifesciences, Irvine, CA, USA) Analgesia: remifentanil perfusion Deep neuromuscular block Active heating with thermal fluid heater and thermal blanket No nasogastric tube Prophylaxis of postoperative nausea and vomiting following Apfel scale Multimodal postoperative analgesia: laparoscopic-guided transversus abdominis plane (TAP) blockade (bupivacaine 0.25%) + intravenous analgesia. Immediate postoperative period Maintenance of FiO2 50% for 2 h after surgery Incentive spirometry In case of atelectasis or hypoxemia, start noninvasive mechanical ventilation (IPAP 12, EPAP4). CPAP in all patients using it previously Avoid morphic drugs Oral fluids 6 h after surgery Sit patient in seat 6 h after surgery Thromboprophylaxis
Postoperative day 1	Liquid diet Active mobilization Start oral analgesia Analytic evaluation of C reactive protein and/or procalcitonin Eventual hospital discharge
Discharge and follow-up	Discharge criteria: No surgical complications, no fever, pain controlled with oral analgesia, full deambulation, patient acceptance Recommendations at discharge Maintenance of thromboprophylaxis for 28 d after surgery Telephone monitoring for 48 h First outpatient visit 15 days after discharge Nutritional recommendations: liquid hypocaloric hyperproteic diet; divided doses

ERAS = enhanced recovery after surgery group; CPAP = continuous positive airway pressure; SAHS = sleep apnea-hypopnea syndrome; IPAP 12 = inspiratory positive airway pressure 12; EPAP4 = expiratory positive airway pressure 4.

**Table 2 ijerph-19-14946-t002:** Distribution of pathologies within the personal history between groups. (Chi-square test).

	ERAS	Standard Care	*p*
Diabetes mellitus	23.8%	33.3%	0.469
Hypertension	35.7%	40.5%	0.822
Dyslipidemia	35.7	30.9%	0.817
SAHS	61.9%	64.3%	0.821
Liver steatosis	73.8%	61.9%	0.238
GERD	14.3%	19%	0.771

SAHS: Sleep apnea-hypopnea syndrome. GERD: Gastroesophageal reflux disease.

**Table 3 ijerph-19-14946-t003:** Baseline anthropometric measurements and weight loss obtained with the balanced preoperative diet 3 months before surgery. (Student’s *t* test).

	ERAS	Standard Care	*p*
Height (cm)	165.6 ± 10.4	163 ± 8.4	0.218
Baseline weight (Kg)	123.3 ± 26.2	122.6 ± +19.4	0.884
Baseline BMI (Kg/m^2^)	44.8 ± 6.6	46 ± 5.9	0.395
Weight after diet (Kg)	115.8 ± 28.1	117.4 ± 18	0.745
BMI after diet (Kg/m^2^)	43.2 ± 6.7	44 ± 4.6	0.526
Total weight loss after diet (%)	6.1 ± 1.9	4.2 ± 1.4	0.688

**Table 4 ijerph-19-14946-t004:** Distribution of pain between groups. (Student’s *t*-test).

	ERAS	Standard Care	*p*
Pain at 6 h (mm)	16.7 ± 17.8	37.1 ± 23.5	<0.001
Pain at 24 h (mm)	19.5 ± 15.8	29.8 ± 18.4	0.008

**Table 5 ijerph-19-14946-t005:** Distribution of costs between groups. (Student’s *t*-test).

	ERAS	Standard Care	*p*	Mean Cost Difference between Protocols
Pharmacologic cost$	337.6 ± 14.9	356.1 ± 16.7	<0.001	18.5
Cost of surgical material and time$	3270.69 ± 322.9	3257.67 ± 277.63	0.843	−13.02
Cost of complementary tests$	311.5 ± 39.6	423.4 ± 58.6	<0.001	111.9
Cost of bed occupancy$	1466.7 ± 1201.6	2828.3 ± 1429.2	<0.001	1361.6
Total cost of the procedure$	5406.86 ± 1334.54	6865.49 ± 1443.24	<0.001	1458.63

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
