# Peer review of "Economic Impact of the Implementation of an Enhanced Recovery after Surgery (ERAS) Protocol in a Bariatric Patient Undergoing a Roux-En-Y Gastric Bypass"

_ijerph, 2022, doi:10.3390/ijerph192214946_

Round 1

Reviewer 1 Report

The work presents the results of a clinical comparative study between an ERAS and a standard-of-care protocols. The conclusions have impactful significance on the clinical outcomes.

Some improvements/comments are raised:

1. It would be good to include in the abstract the equivalence of mean saving of 1458.62$ in precents out of the usual costs.

2. What is the meaning of 37 mm of pain quantification? a description of the scale would be good for the reader. Why different threshold were used for deciding the patients selectred in the two groups (37 mm - standardized care, 16 mm - ERAS protocol)?

3. Why the details on the surgical technique are important? 

4. How the pathologies from Table 2 could have affected the length of stay in PACU of those patients, and implicitly the results?

5. Based on what where the costs decided?

Finally, the paper is very well written and supported by the analysis. 

Additional small improvements could be done:

Abstract, lines 22-24, 27-29: the word costs appears several times in the same sentence. 

Table 1: change peroperative with perioperative

Advice: make sentences shorter (eg: Intro lines 51-54 o rother throughout the text)

Author Response

Thank you for your valuable comments and recommendations.

  1. It would be good to include in the abstract the equivalence of mean saving of 1458.62$ in precents out of the usual costs.

ANSWER: It has been added in the Abstract and in the Results section (Paragraph 3.4.5) “The implementation of an ERAS protocol implied a mean saving of 21.25% of the total cost of the procedure.”

  1. What is the meaning of 37 mmof pain quantification? a description of the scale would be good for the reader. Why different threshold were used for deciding the patients selectred in the two groups (37 mm - standardized care, 16 mm - ERAS protocol)?

ANSWER: The description of the scale has been added “This pain quantification was measured by visual analogic scale (VAS), ranging from 0 (absence of pain) to 100 (unbearable pain).” The different threshold for both groups was established based on the results obtained in our previous study. (Material & Methods, 3rd paragraph)

  1. Why the details on the surgical technique are important? 

ANSWER: We consider the details of the surgical technique relevant, in order to justify the surgical material employed and consequently calculate the costs associated with surgical material and even with the operation time. The surgical procedure of RYGB is not standardized between different surgical groups.

  1. How the pathologies from Table 2 could have affected the length of stay in PACU of those patients, and implicitly the results?

ANSWER: THe following paragraph has been added as a Limitation of the study “Another potential limitation of the study is that the health care costs associated with the procedure have been established for a morbidly obese population with homogeneous characteristics between the two study groups. However, populations with a higher prevalence of comorbidities associated with obesity would have possibly presented higher rates of postoperative complications and would even have made the implementation of the ERAS protocol more difficult, resulting in delayed hospital discharges. Therefore, extrapolation of these results to any type of patient should be taken with caution.” (Discussion, last paragraph).

  1. Based on what where the costs decided?

ANSWER: “The economic costs were calculated according to the scales established for the center where the surgeries were performed. ” (Section 2.6, 2nd paragraph)

Additional small improvements could be done:

Abstract, lines 22-24, 27-29: the word costs appears several times in the same sentence. 

ANSWER: Some words “cost” have been changed to “expenses, price or expenditures”

Table 1: change peroperative with perioperative

ANSWER: It has been changed

Advice: make sentences shorter (eg: Intro lines 51-54 o rother throughout the text)

ANSWER: It has been modified

Reviewer 2 Report

The authors have indeed presented a well-documented and meticulously conducted clinical trial comparing enhanced recovery after surgery (ERAS) and standard protocols for Roux-En-Y gastric bypass surgery. The importance of ERAS protocol has been clearly elucidated by their results. While there are not many comments to improvise or revise the manuscript, there are a few minor issues which could be addressed by the authors. They are as below:

  1. Introduce abbreviations at first use in the abstract.
  2. In the introduction – Lines 71-72, kindly mention that the report refers to an earlier reported study in 2019. For the lay reader, it might to lead to an assumption that the current study is the one being referred to.
  3. Although the study employed a prospective design, why was it non-randomized? Considering the fact that there was no statistically significant difference between the test and control groups in terms of demographic characteristics and underlying comorbidities, the study would have been much stronger if it was an RCT.
  4. The visual analog scale mentioned in your study quantifies pain in millimeters. It would be beneficial if a graphical description of the same is included to the manuscript.
  5. In Table 5, the cost benefit per individual item could be better highlighted by adding a column showing the cost difference between the 2 surgical protocols
  6. While the present study implores the idea of ERAS protocol for bariatric surgery in the setting of a developing to developed economy, Mexico in this instance, how would the same fare in lesser developed or third world economies? The idea of reduced cost and early recovery are even more imperative for countries, which are already strained by a low to meagre healthcare budget and at the same time have higher incidence of obesity disorders, for example BRICS countries. The same could be added as a note in the discussion.

Author Response

We appreciate your valuable comments and recommendations.

Introduce abbreviations at first use in the abstract.

ANSWER: The acronym ERAS has been described at first use in the Abstract (line 15)

In the introduction – Lines 71-72, kindly mention that the report refers to an earlier reported study in 2019. For the lay reader, it might to lead to an assumption that the current study is the one being referred to.

ANSWER: It has been specified that the referred study was conducted in 2019 “Our group conducted the first prospective randomized clinical trial in 2019, comparing the implementation of an ERAS protocol with a standard care protocol in RYGB. This study concluded that the implementation of the ERAS protocol was associated with less postoperative pain, nausea and vomiting, lower levels of analytical acute phase reactants, and earlier hospital discharge, but with similar rates of complications, reoperation, mortality and readmission.”. (Introduction, 5th paragraph)

Although the study employed a prospective design, why was it non-randomized? Considering the fact that there was no statistically significant difference between the test and control groups in terms of demographic characteristics and underlying comorbidities, the study would have been much stronger if it was an RCT.

ANSWER: We totally agree with your appreciation, but the unavailability of Clearsight devices for the calculation of goal-directed intraoperative fluid therapy for some time during the study, prevented uso f randomizing the study. This has been highlighted in the DIscussion as a limitation of the study. “Finally, we would like to highlight as the main limitation of this study, the impossibility of randomizing patients due to the unavailability of Clearsight devices, which would allow the calculation of goal-directed intraoperative fluid therapy for some time during the study.” (Discussion, 13th paragraph)

The visual analog scale mentioned in your study quantifies pain in millimeters. It would be beneficial if a graphical description of the same is included to the manuscript.

ANSWER: The description of the scale has been added “This pain quantification was measured by visual analogic scale (VAS), ranging from 0 (absence of pain) to 100 (unbearable pain).” (Material & Methods, 3rd paragraph)

In Table 5, the cost benefit per individual item could be better highlighted by adding a column showing the cost difference between the 2 surgical protocols

ANSWER: A column with mean cost difference has been added in Table 5.

While the present study implores the idea of ERAS protocol for bariatric surgery in the setting of a developing to developed economy, Mexico in this instance, how would the same fare in lesser developed or third world economies? The idea of reduced cost and early recovery are even more imperative for countries, which are already strained by a low to meagre healthcare budget and at the same time have higher incidence of obesity disorders, for example BRICS countries. The same could be added as a note in the discussion.

ANSWER: The following paragraph has been added in the Discussion “The present study implores the idea of ERAS protocol for bariatric surgery in the setting of a developing to developed economy. However, the idea of reduced cost and early recovery are even more imperative for countries, which are already strained by a low to meagre healthcare budget and at the same time have higher incidence of obesity disorders. The current evidence is based on studies carried out in developed countries, but given this clear economic benefit, it is logical to think that its application in developing countries would also be a cost-effective measure. It is true that the availability of some complex technological devices may be more difficult in this environment, but even without them, many of the items in the ERAS protocols could be implemented without problems, and a clear benefit could be obtained. Further studies should evaluate this hypothesis.” (Discussion, 12th paragraph)

Reviewer 3 Report

Dear Authors,

congratulations for your paper and the quality of your work. 

I would like to suggest only direct references to ERABS guidelines, since have not been mentioned and they're specific for bariatric surgery.

Anyway, the overall quality of your paper is ok.

Best regards

Author Response

Dear Authors,

congratulations for your paper and the quality of your work. 

I would like to suggest only direct references to ERABS guidelines, since have not been mentioned and they're specific for bariatric surgery.

Anyway, the overall quality of your paper is ok.

Best regards

ANSWER: Thank you very much for your appreciations. The ERABS guidelines have been mentioned and References 16 and 17 have been added. (Introduction, 8th paragraph)

Reviewer 4 Report

In the manuscript entitled “Economics Impact of the Implementation of an Enhanced Recovery after Surgery (ERAS) Protocol in a Bariatric Patient Undergoing a Roux-En-Y Gastric Bypass”, the author compared the economic cost of performing laparoscopic Roux-en-Y gastric bypass following an ERAS protocol, with the costs of following a standard-of-care protocol and concluded that the postoperative pain, nausea or vomiting, hospital stay and the costs related to the hospital stay were significantly lower in the ERAS group. The results of this manuscript would be beneficial for evaluating the economic impact of the implementation of an EREAS protocol. I have some minor comments shown below to help improve the clarity of this manuscript. 

1)It would be better to include baseline glucose, insulin levels, and HOMA-IR of the patients as an index for insulin resistance in Table 2. 

2) It would be better to include the lean mass and fat mass changes of the patients in Table 3. 

3) It would be better to show the statistical analyses and p-value for each table.

Author Response

We appreciate your valuable comments and recommendations.

1) It would be better to include baseline glucose, insulin levels, and HOMA-IR of the patients as an index for insulin resistance in Table 2. 

ANSWER: I completely agree with your comment. But unfortunately these variables were not recorded during the study. We have added this as a limitation of the study. “Another potential limitation of the study is that the health care costs associated with the procedure have been established for a morbidly obese population with homogeneous characteristics between the two study groups. However, populations with a higher prevalence of comorbidities associated with obesity would have possibly presented higher rates of postoperative complications and would even have made the implementation of the ERAS protocol more difficult, resulting in delayed hospital discharges. For instance, in our study only the prevalence of diabetes mellitus diagnoses was recorded, whereas baseline glucose, insulin levels, and HOMA-IR of the patients as an index for insulin resistance would have added more information and could even be markers of patients more prone to develop complications or presenting difficulties in the adherence to the ERAS protocol. Body composition, as measured by bioimpedancy, including lean and fat mass percentages would have also added a valuable information. Therefore, extrapolation of the results of the present study to any type of patient should be taken with caution. ” (Discussion, last paragraph)

2) It would be better to include the lean mass and fat mass changes of the patients in Table 3. 

ANSWER: Similarly to the previous point, bioimpedancy was not carried out. (Discussion, last paragraph)

3) It would be better to show the statistical analyses and p-value for each table.

ANSWER: Statistical tests used and p values have been added in each table.